# An Efficient and Frequency-Scalable Algorithm for the Evaluation of Relative Permittivity Based on a Reference Data Set and a Microstrip Ring Resonator

**DOI:** 10.3390/s22155591

**Published:** 2022-07-26

**Authors:** Miroslav Joler

**Affiliations:** Department of Computer Engineering, Faculty of Engineering, University of Rijeka, 51000 Rijeka, Croatia; mjoler@riteh.hr; Tel.: +385-51-651-462

**Keywords:** evaluation of relative permittivity, full-wave solvers, microstrip ring resonator, Schumaker spline, dielectric characterization

## Abstract

In this paper, a fast and efficient algorithm for the evaluation of relative permittivity of a solid dielectric sample, when measured by a microstrip ring resonator, is proposed. It is verified for permittivity values up to 10 and material-under-test thicknesses up to 8 mm, which cover a wide range of prospective materials that may be used in electronics and communications. The algorithm was tested on 11 samples of various permittivity values and thicknesses and showed a very good agreement with their nominal permittivity values. The maximum error was within 10% even for the sample thicker than 7 mm, while the results for the four standard laminates (TLX8-060, RF60A-0300, RF60A-0620, and FR4) showed an average error of 2.34%. Attractive features of the proposed algorithm are that the results contained in the reference set are frequency-scalable, applicable to many pairs of unknown permittivity and sample thickness values, unbiased, and easily appendable with additional reference points if higher accuracy is sought.

## 1. Introduction

The design of radio frequency (RF) and microwave circuits requires sufficiently accurate information on the relative permittivity ($\epsilon_{r}$) of the materials that are used in the design because relative permittivity has a significant impact on the accurate design of a circuit, especially on the dimensions of the circuit or its resonant frequency. Besides the permittivity, the thickness of the material also affects the results. These parameters are included in the closed-form design expressions (e.g., see [1,2,3,4]) and data sheets [5,6,7,8]; their impacts are nowadays typically verified by full-wave electromagnetic solvers.

However, in the design of modern circuits, especially innovative circuit designs for wearable electronics, circuits will occasionally comprise a nonstandard substrate (or superstrate) materials, such as fabrics (e.g., in wearable antennas), whose relative permittivities are not readily known due to the fact that relative permittivity is not a relevant parameter for textile-related communities and, therefore, has not been measured and specified as part of the standard data sheets that pertain to traditional fabrics. It is, then, up to researchers to overcome that lack of information by performing a proper evaluation of relative permittivity as the most impactful property of the material for the RF/microwave circuit design.

In that sense, a number of methods have been proposed over time [9], each with its advantages and disadvantages. In [10,11], interested readers can find recent overviews of the methods that are offered by prominent commercial entities, while in [3,4], one can find a more detailed approach to a variety of proposed methods. Clearly, the established methods offer solutions from low frequencies (e.g., the parallel plate method), over mid-range frequencies (e.g., the resonant cavity method), to wide bandwidth measurements from relatively low frequencies to medium- (e.g., the coaxial probe or the transmission line method), or even millimeter-wave frequencies (such as the free-space technique) [10]. It is up to a user to identify which method best suits his/her case. Other aspects of the user’s choice include how destructive the method is toward the sample under test (SUT), necessity (or not) for a wideband characterization of the material to be tested, a physical state of the SUT, etc. One common trait of all of them is that the determination of permittivity is a subtle task with inherent sensitivity to the measurement process and the required precision regarding the preparation of the SUT in terms of its size and placement on/in the measurement fixture (if there is one). Overviews of various methods can be further found in [12,13].

Among the papers that tackled a specific measurement technique, techniques and algorithms based on the use of the *transmission line* have been a popular topic of research [14,15,16,17,18,19]. Various versions and applications based on the *microstrip ring resonator* (MRR) configuration were covered in [20,21,22,23,24,25]. Fewer papers have focused on the use of the parallel plate method, mostly for low-frequency applications, e.g., [26]. On the other hand, the *waveguide* is an appealing structure for wideband measurements [27], but it requires a very precise preparation of the measurement sample size and its precise placement in the slot within a waveguide, which is one reason why it has been relatively less discussed in comparison with the *transmission line method*, which has been the focus of various proposed methods. *Resonant cavity methods* generally have good accuracy for a single frequency and are nondestructive for a material-under-test, but there are differences in the particular circuit design to perform the measurements. Some circuits are based on the open microstrip structure (such as an MRR), whose upper surface serves as a testbed to place a sample of a material to be measured [24], while some others (supposedly more accurate once) are based on the split post dielectric resonator fixture to serve as the testbed for a SUT [28]. Additionally, an MRR is sometimes designed on the substrate whose permittivity is, in fact, the object of the measurement and permittivity evaluation [21], as opposed to various other cases where a SUT is placed on top of the resonator surface in order to be measured. In [29], a split ring resonator (SRR) is used to retrieve electromagnetic parameters from inhomogeneous metamaterial. The measurement method with an *open coaxial probe* is suitable to determine the permittivity of liquid dielectrics [30].

Interestingly, the bulk of the various proposed methods can be traced back to the 1960s–1980s, with fewer papers in the 1990s and later, yet characterization of dielectric properties remains quite a subtle task to perform. It is apparently due to inherent challenges that are involved in the measurement procedures, apparatus, preparation, and placement (or insertion) of the sample to be tested. Moreover, it can be attributed to a nonlinear nature of the spectral response of a SUT, as well as the numerical (i.e., algorithmic) part of the process, which follows after the measurements are conducted and requires a certain algorithm to post-process the results of the measurements.

Due to all of that, commercial kits are still quite costly, while offering various (of the aforementioned) techniques with corresponding measurement probes and accessories [31,32,33]. As such, they are not quite affordable to labs and researchers that work on modest budgets. On top of it, even acquiring some of the leading commercial solutions still does not guarantee a high accuracy in all cases and has limitations in sample thickness or volume and permittivity value range, and there are other disclaimers pertaining to tolerances and measurement accuracies. Due to that, researchers who do not need to characterize unknown dielectrics so frequently (to feel that procurement of very expensive instrumentation is justified) are destined to conduct their own design and manufacturing of a suitable measurement setup, leading to results that are good enough for the practical needs.

In our case, it was adequate to cling to a measurement circuit that was simple enough to design and cost-effective to manufacture, with the prospect of providing accurate enough measurement results at discrete frequencies of our choosing (e.g., relative error under 10%) and suitable for SUTs with lower permittivity values (e.g., $\epsilon_{r}\leq 10$) and modest sample thicknesses of up to a few millimeters. Taking into account the traits of the aforementioned methods—an MRR, as a representative of the resonant methods, was chosen, mainly for its simplicity, affordability, and accuracy.

While in a recent paper [34] we discussed a simplified approach to determining relative permittivity using an MRR with a variational method-based algorithm, in this paper, we propose a faster, efficient, and frequency-scalable algorithm to accurately compute the relative permittivity value of an unknown low-loss material for any value of its thickness, from a submillimeter thickness up to 8 mm. As its input values, the algorithm only needs the SUT thickness (*S*), and the two measured resonant frequencies—the one without a SUT on the MRR surface, which is commonly referred to as the “unloaded” resonant frequency, $F_{0}$, and the one with a SUT on the MRR surface, which is referred to as the “loaded” resonant frequency ($F_{1}$). To evaluate the unknown permittivity of the given SUT, the algorithm then uses the *pre-stored reference data set* along with those three input variables. In the following sections, the computational benefits of this approach will be compared with the variational method-based algorithm in terms of the algorithm complexity, the computational time, and accuracy.

The paper is structured as follows. In Section 2, the key aspects of a variational method-based algorithm (VMA) are outlined in order to later serve for comparison with the algorithm proposed in this text. The proposed algorithm is then described in Section 3, followed by testing of the algorithm performance using 11 benchmark samples in Section 4. The flexibility of the approach to append the reference data set with additional data points, as well as an inherent frequency scaling of the reference data set is further discussed in Section 5. The appealing characteristics of the algorithm are discussed in Section 6, and include a discussion on the means to further improve the accuracy. Lastly, Section 7 summarizes the paper.

## 2. A Brief Reference to the Previous Procedure

Our evaluation of a SUT permittivity value was previously based on the procedure described in [34] in detail; as for the purpose of this paper, only its essentials are reviewed. The variational method-based algorithm comprises three key computational steps before the unknown permittivity can be evaluated.

First—the effective permittivity of the unloaded MRR, $\epsilon_{f0}$, is computed by
(1)$$\epsilon_{f0}=\frac{C_{1}}{C_{0}}$$
where $C_{0}$ is the capacitance of the unloaded MRR when air is used as the MRR substrate, while $C_{1}$ is the capacitance of the unloaded MRR when the particular substrate is used in the design of the MRR, e.g., FR4. (Note: the term *unloaded* is used for the MRR without the SUT placed on its surface, while the term *loaded* is used for the MRR having a SUT placed on its surface.)

Second—the resonant frequencies of the unloaded- and loaded-MRRs must be measured using a network analyzer (NA) in order to compute the effective permittivity of the loaded MRR, $\epsilon_{f1}$. When a SUT is placed on the MRR surface, a downshift in the resonant frequency occurs due to the presence of non-air material in the overall MRR structure. The effective permittivity of the loaded MRR, $\epsilon_{f1}$, is then equal to
(2)$$\epsilon_{f1}=\epsilon_{f0}\;{\left(\frac{F_{0}}{F_{1}}\right)}^{2}$$
where $F_{0}$ and $F_{1}$ are the resonant frequencies of the *unloaded*- and *loaded*-MRRs, respectively, and $\epsilon_{f0}$ has already been computed by (1). This result serves in the next step as the specific value of the effective permittivity that the relative permittivity will be evaluated for.

Third—to find the particular value of the relative permittivity of the SUT, $\epsilon_{2}$, a relationship between the effective permittivity of the loaded MRR, $\epsilon_{f}$, and any value of $\epsilon_{2}$ needs to established first, which is done by varying the value of $\epsilon_{2}$ from 1 to some arbitrary value $\epsilon_{2_{max}}$ and computing $\epsilon_{f}$ by
(3)$$\epsilon_{f}=\frac{C(\epsilon_{2})}{C_{0}}$$
where *C* is the respective capacitance for the specific $\epsilon_{2}$ value in the vector of $\epsilon_{2}$ values. When such a functional dependence is established as $\epsilon_{f}=\xi \left(\epsilon_{2}\right)$ (or vice versa, i.e., $\epsilon_{2}=\xi^{-1}\left(\epsilon_{f}\right))$, graphically or numerically, the particular $\epsilon_{2}$ is readily obtained as the counterpart of the specific value of $\epsilon_{f}=\epsilon_{f1}$, where $\epsilon_{f1}$ was computed by (2) in the second step.

To summarize, this approach involves a computation of the capacitance of the air-based MRR ($C_{0}$), the capacitance of the unloaded MRR filled with a particular dielectric substrate ($C_{1}$), such as FR4, then the computations of $\epsilon_{f0}$, $\epsilon_{f1}$, and a vector of $\epsilon_{f}$ based on the arbitrary vector of $\epsilon_{2}$ values, as defined by (3). It is important to note that the computation of any of the above capacitances in expressions (1), (2), and (3) involve a respective evaluation of the integral function [34]
(4)$$\frac{1}{C}=\frac{1}{\pi \;Q^{2}\;\epsilon_{0}}\int_{0}^{\infty }{\left[\tilde{f}\left(\beta \right)\right]}^{2}\;\tilde{g}\left(\beta \right)\;\tilde{h}\left(\beta \right)\;d\beta$$
each time with a different set of the input parameter values, which are pertinent to the particular case, i.e., respective input parameters pertaining to either $C_{0}$, $C_{1}$, or *C*, where
(5)$$\begin{array}{c} \frac{\tilde{f}\left(\beta \right)}{Q}=\frac{8}{5}\left\{\frac{\sin(\beta w/2)}{\beta w/2}\right\}+\frac{12}{5\;{\left(\beta w/2\right)}^{2}}\\ \cdot \left\{\cos\left(\beta w/2\right)-\frac{2\sin(\beta w/2)}{\beta w/2}+\frac{\sin^{2}\left(\beta w/4\right)}{{\left(\beta w/4\right)}^{2}}\right\} \end{array}$$
with $\epsilon_{0}=8.854\cdot 10^{-12}\;\frac{F}{m}$ being the free-space permittivity and *w* the width of the ring and the feed line, while $\tilde{g}\left(\beta \right)$ is given by
(6)$$\begin{array}{c} \tilde{g}\left(\beta \right)=\frac{\epsilon_{3}\;d+\epsilon_{2}\;s}{\left|\beta |\;\{\;\epsilon_{3}\;d\;\left[\;\epsilon_{1}\;h+\epsilon_{2}\;s\right]+\epsilon_{2}\;\left[\;\epsilon_{2}+\epsilon_{1}\;h\;s\right]\right\}} \end{array}$$
where
(7)d=coth(|β|D)
(8)s=coth(|β|S)
(9)h=coth(|β|H)
and
(10)$$\tilde{h}\left(\beta \right)=\frac{1}{2}\left\{1+\frac{\sin h(|\beta |d-|\beta |t)}{\sin h(|\beta |d)}\right\}$$
with *S* being the SUT thickness, *D* being the cover layer thickness (as discussed in [24,35,36]), and *t* being the copper layer line thickness. Conveniently approximating t→0, $\tilde{h}\left(\beta \right)\to 1$ in (4).

## 3. The Proposed Algorithm Based on the Reference Data Set

Having experienced the variational method-based algorithm to determine the $\epsilon_{2}$ of the SUT [35,36], we contemplated an alternative algorithm that would be simpler, faster, and would efficiently compute a relative permittivity of a SUT for *any* SUT permittivity value and *any* thickness value within a realistic range of interest, which is approximately $\epsilon_{r}<10$ and S<8mm. To achieve that, it was required to establish the dependence of $F_{1}$ on the SUT thickness *S* and permittivity $\epsilon_{r}$ of the prepared referential data set and then apply a data fitness algorithm that closely followed the data. A particular SUT permittivity ($\epsilon_{2}$) could then be evaluated, with a comparable accuracy faster than it was achieved by using the variational method-based algorithm.

### 3.1. The Approach to Creating a Reference Data Set

The basic idea of the algorithm stemmed from the fact that by taking advantage of the full-wave solver, we could consistently model the unloaded- and loaded-MRR and achieve quite realistic results in terms of the respective resonant frequencies for the given SUT thickness and permittivity. Figure 1 illustrates a computational model of an MRR with a SUT placed over its surface. The advantage of using a computational model to establish the reference data lies in the possibility of having repetitive evaluations with equal treatment of the SUT, without introducing different human errors from evaluation to evaluation. This approach actually substitutes for the measurements that would otherwise be performed in order to establish the reference data set, during which, the operator is prone to prepare and place the samples unequally from measurement to measurement, to measure the SUT with some errors from sample to sample or to use insufficiently accurate information on the SUT permittivity. Within a full-wave solver, the operator can consistently work with exact reference values of interest.

Now, if several thickness values of the same material are computationally analyzed, a vector of $F_{1}$ points is obtained for the respective thickness values *S* with a given relative permittivity $\epsilon_{2}$ of the reference SUT. (Credible values of the SUT thicknesses, *S*, and the permittivity value, $\epsilon_{2}$, for the chosen dielectric laminate, can be obtained from the manufacturer’s data sheet.) The procedure is then repeated for a few other known materials with different permittivity values, to end up with a few series of data points representing the decay of the resonant frequency of the loaded MRR ($F_{1}$) as a function of both *S* and $\epsilon_{2}$, i.e., $F_{1}=\Psi \left(S,\epsilon_{2}\right)$. To make it simpler, the dependence of $F_{1}$ with respect to varying *S* is analyzed first, then followed by the analysis of the dependence of $F_{1}$ with respect to varying $\epsilon_{2}$. Ultimately, should one be able to find a sensitive data-fitting algorithm that would accurately interpolate (and to some extent extrapolate) between the given set of $F_{1}$ points as a function of *S*, one could evaluate $F_{1}$ for *any* thickness of a SUT with some permittivity value, based on just a few reference samples that are stored in the algorithm in advance. That dependence is described by
(11)$$F_{1}=\xi \left(S\right){|}_{\epsilon_{2}}$$
and here referred to as the *horizontal* interpolation.

Following that, a *vertical* interpolation between the reference curves (having different reference permittivities) is then performed for a given *S*, to achieve the resulting SUT permittivity value $\epsilon_{2}$ based on the given $F_{1}$. The dependence with respect to $\epsilon_{2}$ is
(12)$$F_{1}=\chi \left(\epsilon_{2}\right){|}_{S}$$

So, with the two steps of “horizontal” and “vertical” interpolations operating on the reference data points, relative permittivity of unknown material can be efficiently determined from the three measured input values: $F_{0}$, $F_{1}$, and *S*, avoiding repetitive computations of the integral in (4), which is computed with different settings in each of the expressions (1), (2), and (3).

### 3.2. Data Fitting Algorithms

Figure 2 shows the reference data set for the four laminates that were used here as the references (see Table 1), while the uppermost set of constant values represents air as a quasi-sample, which is needed for the computation of samples whose permittivity value is lower than the lowest used permittivity value of the reference data set. The SUT thickness values *S*, listed in Table 1, were based on the respective data sheets, while the results of the respective $F_{1}$ values were achieved by computational modeling in the Feko full-wave solver [37], analogously to the model shown in Figure 1.

As can be noticed, the reference laminates neither have the same number of sample thicknesses nor equal values of their thicknesses. Due to that, the data fitting task encounters unequal lengths of the respective vectors and unequal values of the independent variable *S*. There is also an unequal trend of the $F_{1}$ value decay, as can be seen in Figure 2. It was also realized that one point had to be common to all the data sets and that point is the resonant frequency of the unloaded MRR, $F_{0}$ (i.e., the frequency value for S=0).

### 3.3. A Fitting Attempt with Exponential Functions

At first glance, the curves appeared to follow an exponential decay of $F_{1}$ values and we tried to fit it with exponential terms of the form exp(−αS), where α was tested with various values, but that did not result in good data fitting because only one half of any curve could be fitted reasonably well, while the other half exhibited poor fitting. It can be observed in Figure 3 where, for α=0.028, better fitting was obtained only for the last part of the curve, as shown in Figure 3a. In contrast, using α=0.06, better fitting was obtained for the first part of the curve, as illustrated in Figure 3b. Similarly, no particular exponential coefficient α would enable good fitting over the entire curve. An alternative idea was to use a different exponential coefficient for each half of the given data set. That, on the one hand, still does not guarantee close fitting for either segment of the curve. On the other hand, it complicates the fitting process because every reference curve has its own trend of decay. With that, one would have to define at least two exponential fitting coefficients for each reference curve and such an approach would be both clumsy and not guarantee a good fitting.

### 3.4. Polynomial Fitting Attempt

Another *classical* approach involved making use of a *polynomial* fitting function, but that did not achieve a close-enough fitting either. As it is known about polynomial fitting, it is important to pick an *optimal* order of the polynomial for the given data set; no particular order produced satisfactory results in this case. The fitting curves fluctuated around the given values, rather than passing through them. In addition, the polynomial fit exhibits stronger divergence from the given data values near the boundaries of the data set, making the approximation even worse.

### 3.5. Shape-Preserving Algorithm for the “Horizontal” Data Fitting

Ultimately, it was recalled that we had implemented a superior data-fitting algorithm in one earlier publication [38]. The algorithm is referred to as a *shape-preserving* algorithm (SPA). It is characterized by accurately following the given data points, no matter how challenging their trend of values may be and it is superior to other fitting algorithms. While we previously implemented it in MATLAB programming language, we now implement it in Julia programming language [39], specifically via the *Schumaker Spline shape-preserving interpolation algorithm* [40], which not only closely follows the data pattern and interpolated between the data points, but also extrapolates the data vector very logically on either side outside of the given data set (Figure 4). One exception to that is the curve for RF-35, but not due to the SPA itself, but due to the fact that the data vector did not have values beyond S=1.52mm, and was therefore “lacking” the $F_{1}$ result around S=3mm, such as the other reference vectors. Due to that, the extrapolated curve in that segment does not follow the decay trend analogously to the other curves; instead, it decays monotonously linearly. This can be mended by running simulations with additional values of *S*, which will be shown later in the text in Section 5.1.

In Figure 4, the dots show the original reference data points of $F_{1}=\xi \left(S\right){|}_{\epsilon_{2}}$ and the solid lines show the Schumaker shape-preserving spline-interpolated results.

### 3.6. The “Vertical” Interpolation of the Data (between the Reference Curves)

Having accomplished the *horizontal* interpolation of the reference data, it remained to define a *vertical* interpolation between the curves shown in Figure 4, which would ultimately enable achieving accurate results for *any* value of the SUT permittivity that falls between the reference curves.

In this step, the sample thickness *S* is a given parameter, along with the measured value of $F_{1}$. As the curves exhibit different vertical separations for different values of *S* and different trends of their respective decays, the idea was that the resulting curve, which will lie somewhere between the two adjacent curves, proportionally adopts as much property of the reference curve that it is closer to and the remaining property of the other adjacent reference curve it is further from. It is understood that only two reference curves will bound any pair of $\left(S,F_{1}\right)$ values—one reference curve as the upper bound and the other reference curve as the lower bound. Once the two bounding curves are identified, we analyze such a vertical interval between the upper reference curve, the given point $\left(S,F_{1}\right)$, and the lower reference curve, and calculate how relatively close the given point $\left(S,F_{1}\right)$ is to either of the curves. Figure 5 illustrates one such vertical interval between the boundary points *a* and *b*, except for being shown with a horizontal orientation. The unknown permittivity value $\epsilon_{x}$ to be found is $p_{1}$ far from *a* and $p_{2}$ far from *b* (Appendix A).

Based on that, the resulting curve will attain the calculated percentage of the upper curve (*a*) and the remaining percentage of the lower curve (*b*), which is inversely proportional to the distance that the $\left(S,F_{1}\right)$-point has to each curve. That is algorithmically defined by: (13)$$\begin{array}{cc} a & =\epsilon_{r}\left[i\right] \end{array}$$(14)$$\begin{array}{cc} b & =\epsilon_{r}\left[i+1\right] \end{array}$$(15)$$\begin{array}{cc} p_{1} & =\left(\epsilon_{x}-a\right)/\left(b-a\right) \end{array}$$(16)$$\begin{array}{cc} p_{2} & =1-p_{1} \end{array}$$
where *a* is the permittivity value of the upper curve in the observed interval (with index *i*), *b* is the permittivity value of the lower curve in the observed interval (with index i+1), while the proportionality factors $p_{1}$ and $p_{2}$ represent the proximity of the given SUT pair of values $\left(S,F_{1}\right)$, where $p_{1}$ is the proximity of the given point to the upper curve, $p_{2}$ is the proximity to the lower curve, and $\epsilon_{x}$ is the unknown permittivity of the SUT that is to be computed by this algorithm (previously denoted as $\epsilon_{2}$, while we now denoted it by $\epsilon_{x}$ to emphasize that it is still the unknown permittivity of a SUT that is yet to be evaluated). Following that, the algorithmically estimated resulting curve of the given SUT, designated as $F_{est}$, is established as
(17)$$F_{est}=p_{2}\;F_{vals}\left[i,:\right]+p_{1}\;F_{vals}\left[i+1,:\right]$$
where vectors $F_{vals}\left[i,:\right]$ and $F_{vals}\left[i+1,:\right]$ represent the values of the upper and the lower bounding curves, respectively, and $p_{1}$ and $p_{2}$ are evaluated as in (15) and (16). The resulting curve, $F_{est}$, contains the vector of the loaded resonant frequency, $F_{1}$, and the sought-after value of the SUT permittivity, $\epsilon_{x}$. The ultimate value of $\epsilon_{x}$ is then obtained by scanning $\epsilon_{r}$ from the upper bounding curve towards the lower bounding curve and testing whether for that particular value of $\epsilon_{r}$ the measured value of $F_{1}$ matches the $F_{est}$ value (of the estimated curve) for the given *S*. That scan is performed for the nearest value of $S_{i}$, given the chosen resolution of the SUT thickness vector, and the arbitrarily chosen step size of $\Delta \epsilon_{r}$ during the scanning process of searching for the accurate permittivity value. For example, we initially divided the SUT thickness into 228 intervals, which for the SUT ranged between 0 and 3.5, and gave a resolution of ΔS=0.015mm. The step for scanning the permittivity value was arbitrarily set to $\Delta \epsilon_{r}=0.1$, which sounds reasonable for various practical purposes (e.g., antenna design). It is to be noted that the resulting $F_{est}$ curve changes its particular shape dynamically during the scanning loop due to the varying nature of $p_{1}$, $p_{2}$, and $F_{est}$ as defined by (17). (An animated video clip, showing the process of searching for the exact value of the SUT solution is available in the online version of the article.)

## 4. Benchmark Testing

The proposed algorithm was benchmarked against 11 samples of available sample materials that had already been evaluated within our previous paper [34] that was based on the variational method. Here, the results are compared in Table 2, where $\epsilon_{n}$, $\epsilon_{x}$, and $\epsilon_{b}$ denote the *nominal* value of relative permittivity, the computed value of relative permittivity by the variational method-based algorithm from [34], and the computed value of relative permittivity by the *reference data set-based algorithm* proposed in this paper.

As can be seen, the results obtained by the proposed algorithm (RDSA) are very close to the results obtained by the previously discussed algorithm (VMA) [34] and, more importantly, are in good agreement with the nominal permittivity values ($\epsilon_{n}$) shown in column 3.

Figure 6 shows the computed solution curve (the gray one) when the RDSA was tested for the samples of RF60A-0300, FR4, glass, and Plexiglass 1 laminates, having pairs of $\left(S,F_{1}\right)$ values as given in Table 2. The computed solutions of the relative permittivity, $\epsilon_{b}$, are graphically marked by a red plus sign on each of the four solution curves, while their numerical values can also be found listed in the last column of Table 2. Using this proposed algorithm with the given input values, the evaluated solutions are $\epsilon_{b}=6.0,\;4.7,\;6.85,\;\mathrm{and}\;2.5$, respectively.

## 5. Flexibility and Frequency Scalability of the Reference Data Set

In this section, we present two helpful traits of this approach. One of them concerns an easy extension to the initial reference data set where it seems prudent to improve the accuracy of the interpolation process and, consequently, the accuracy of the final $\epsilon_{x}$ value. The initial set can be appended either by additional points within the same reference curve, as illustrated in Section 5.1, or by a whole new vector of reference points (i.e., a new curve).

Another presented feature is the *frequency scalability* of the initial reference data set, which is a very significant trait to make the approach a lot more applicable to various manufactured MRRs and their $F_{0}$ values, and not just in the case of the MRR used here.

### 5.1. Extending the Reference Data for the Thicker Samples

So far, in preparation of the reference data set, we followed the specific genuine data that were affiliated with the particular laminate, as specified in Table 1, and it is evident that for these materials, the sample thickness ends at 3.18 mm. This speaks to the fact that this range of dielectric thicknesses is quite realistic and sufficient when it comes to the design of the most popular PCB circuits, but if one wanted to extend the utility of the method to thicker samples, the initial reference data thickness does not suffice and should be extended. Hence, we appended the initial data sets with additional thickness points up to the maximum thickness of 7.00 mm, with equidistant sample separation of ΔS = 1 mm in most cases, according to Table 3.

Such an extended data set, as illustrated in Figure 7, consequently enabled the data fitness in a wider range of the SUT thicknesses, with the last reference points having S=7mm, and the extrapolation algorithm was set to extrapolate the curves up to S=7.5mm in this case. The extended data set enabled us to compute the relative permittivities of the three thicker samples, which were Plexiglass 2, Plexiglass 3, and polytetrafluoroethylene (PTFE), with their input values given in Table 4.

For comparison, the last two columns are the results of the relative permittivity computed by the initial reference data set, $\epsilon_{b}$, (i.e., up to S=3.18mm) and the extended reference data set, $\epsilon_{e}$, (i.e., up to S=7.00mm). It can be observed that the result in the last column (i.e., $\epsilon_{e}$) is closer to the nominal value of PTFE (i.e., lesser error!), while the results for the Plexiglass 2 and Plexiglass 3 samples are also within the range of the expected result and probably more accurate than $\epsilon_{b}$, which was computed with the original (i.e., shorter) data set. Figure 8a illustrates how the interpolation of the data up to 7 mm using the reference set that stretches only up to 3.18 mm does not follow the natural decay of dielectric laminates. It is most significantly observed with the blue curve (i.e., RF-35), but is also very clear for the two curves below it (i.e., RF-60A and CER-10). In contrast, the interpolations for the extended data sets shown in Figure 8b do have a natural decay trend and, thus, can be more trusted in terms of the result.

Together with Figure 8b, the results in Figure 9 show the graphical solutions for the other two samples from Table 3. A careful reader will notice that $F_{1}$ shows equal values for both the Plexiglass 2 and Plexiglass 3 sample in Table 3, which is typically not expected for the sample of different thicknesses, but that fact leads us to an observation that the downshift of $F_{1}$ becomes less responsive, if at all when it comes to thicker samples. That is actually one of the general characteristics of measurements by an MRR—to become less sensitive for thicker samples or materials with higher dielectric loss.

### 5.2. Frequency Scaling of the Reference Data

Another convenience of this algorithm is the *frequency scaling* of the reference data set. Namely, although the reference data set was computationally generated with respect to the particular simulated unloaded resonant frequency, $F_{0s}$, the decay behavior of $F_{1}$ is independent of the particular $F_{0}$. The decay primarily depends on the dielectric permittivity *r* and the SUT thickness *S*, no matter what the $F_{0}$ value is (which is the upper leftmost point in the graphs, where S=0mm, and is common to all the curves). Due to that, the algorithm is scalable to higher as well as lower $F_{0}$’s, which is conducted in a straightforward way. The reference data points are first normalized to the unloaded resonant frequency of the computational model, $F_{0s}$, to obtain the normalized coefficients for all of the curves. Equation (18) shows it—where $F_{1s}$ is the vector of the loaded resonant frequencies of the computational model and $F_{c}$ are the normalized coefficients. Those normalized coefficients, $F_{c}$, then multiply the actual measured value of the unloaded resonant frequency of the MRR in use, $F_{0m}$, to quickly rescale the initial reference data sets to the particular value $F_{0m}$ of the given MRR, as shown by (19), where $F_{1}$ is then the vector of the rescaled values of the reference set.
(18)$$\begin{array}{cc} F_{c} & =F_{1s}/F_{0s} \end{array}$$
(19)$$\begin{array}{cc} F_{1} & =F_{0m}\cdot F_{c} \end{array}$$

The benchmarking phase of this work effectively proved it to be valid, because $F_{0s}$, which was obtained by the computer simulations ($F_{0s}=2503\;\mathrm{MHz}$), was not identical to $F_{0m}$, which was obtained by the manufactured MRR ($F_{0m}=2506\;\mathrm{MHz}$). In other words, the reference results were prepared computationally with $F_{0s}$, while the measurements of $F_{1}$, using the manufactured MRR, had to be conducted with $F_{0m}$. The frequency scaling of the simulated reference data then came as the logical step to reconcile the simulated values with the values obtained by a particular MRR and enable the applicability of the algorithm independent of the particular MRR $F_{0m}$. The presented results confirm that the approach described above worked. It is viable because starting from the unloaded resonant frequency value, $F_{0}$, the curves retain the same shape and decay trend based on the given SUT $\epsilon_{r}$ and *S*, no matter what the $F_{0}$ value is.

## 6. Discussion on the Characteristics of This Approach

Let us now review the appealing characteristics of this reference data set-based algorithm (RDSA) in comparison to the variational method-based algorithm (VMA) that we utilized in the previous paper [34] and with the prospect of being applicable to other measurement methods.

Besides the measurements of $F_{0}$ and $F_{1}$, which are necessary by either algorithm, the VMA requires multiple calculations of the integral (4). First, two capacitances, $C_{0}$ and $C_{1}$, have to be computed to evaluate (1). Then, vector $C(\epsilon_{2})$ has to be computed to evaluate (3), for which the integral in (4) has to be run *N* times, where *N* is the length of vector $C(\epsilon_{2})$. Although the overall procedure to determine the relative permittivity of some material is, by its nature, a fairly slow process, where a few more seconds do not make much difference, and modern personal computers can conduct the computational part of the process in about 20 s, give or take a few seconds, the VMA is numerically quite demanding, not to mention that in (1), one has to vary the dielectric materials when computing $C_{0}$ and $C_{1}$, while in computing (3), $\epsilon_{2}$ has to be varied within the chosen arbitrary range, which makes the overall calculation subtle.Unlike that, RDSA avoids the computation of the challenging integral. The RDSA algorithm simply takes the given value of the SUT thickness *S*, measures $F_{1}$, and scans through the well-interpolated reference data set, searching for the closest match between *S* (on the abscissa) and $F_{1}$ (on the ordinate). Due to that, the RDSA, with the initial reference data set, takes only about 1/10 of the computation time of the VMA, for the comparable accuracy within 10% of error with respect to the nominal value of the permittivity, based on multiple tests we conducted with the samples presented here. Even when the original reference set was extended, as discussed in Section 5.1, and the number of the interpolation points increased, in order to maintain a fair resolution of the SUT thickness points with the extended data set, the computation did not suffer any noticeable extra time and barely changed, for example, from 0.26 s for the initial data set to 0.43 s for the extended data set, which is a negligible difference for this purpose, by all means.Using a full-wave solver, the preparation of the reference data set was conducted in a consistent and unbiased way by means of 3D computational models, just as was the computation of the loaded resonant frequency for the reference thicknesses. Such a computational approach provides accuracy of the dimensions of the SUTs and their consistent placements on the surface of the MRR, just as it was intended, without introducing random errors from one sample to another.We believe that this algorithm can be used with some other dielectric-characterization methods or for characterization of magnetic materials, in which case each curve would just be represented by a different independent and dependent variable, while the principle of the algorithm would remain the same.

### 6.1. Means to Further Improve the Accuracy

The RDSA algorithm is structured in a way that leaves room for the accuracy of the solution to be further improved, by:(a)Having more reference permittivity curves in order to lessen the vertical separation between the reference curves (in Section 3.6; we referred to it as the “vertical interpolation”).It is easy to prepare a new reference vector for any desired permittivity if the situation for any sample of interest would suggest that the space between any two adjacent curves be lessened for the sake of greater accuracy of the solution. An example is illustrated in Figure 10 where a curve with $\epsilon_{r}=1.2$ (the “low er” cyan curve) is inserted between the air curve and the TLX8 curve ($\epsilon_{r}=2.55$) to eventually improve the accuracy of the solution for the jeans sample, for example. Without this curve, the obtained solution by the RDSA was $\epsilon_{x}=2.0$, while with this added curve, the newly obtained solution, shown by the red plus sign on the grey solution curve, is $\epsilon_{x}=1.9$, which is closer to the quoted nominal value. The result could have possibly been even better if the “lower” curve had been prepared for $\epsilon_{r}=1.4$, for example, thereby further reducing the space between it and the TLX8 curve.(b)Eventually having more data in the horizontal direction, i.e., having more sample thickness points for the reference data set. In this paper, this means having more reference resonant frequency values as a function of the sample thickness (see (11) and the discussion in Section 5.1). With a computational model, it is easy to add more reference points. However, with a small variance of the curve, its benefit is limited.(c)Setting a smaller step in the *vertical interpolation* stage of the algorithm ($\Delta \epsilon_{r}$), but it has its limitations regarding the resulting improvement, depending on the actual quantity of the reference data. In other words, it is not a problem to reduce the step size, but it might not reach any better result for the particular case of interest. For example, when $\Delta \epsilon_{r}$ was reduced from 0.1 to 0.01, the final result on jeans permittivity slightly improved from 1.9 to 1.87, while for the glass sample, it improved from 6.85 to 6.80.(d)Eventually setting a smaller step size ΔS in the *horizontal interpolation*, to make sure the Schumaker-spline data fitting work as well as possible, but in our testing so far, the interpolation performs just as well with a smaller number of the interpolation points along the abscissa, as with an increased number of them. In this case, this was due to the monotonicity of these curves, while in the case of some other reference data set (i.e., by some other method), reducing ΔS might have a more significant impact.

**Figure 10 sensors-22-05591-f010:**
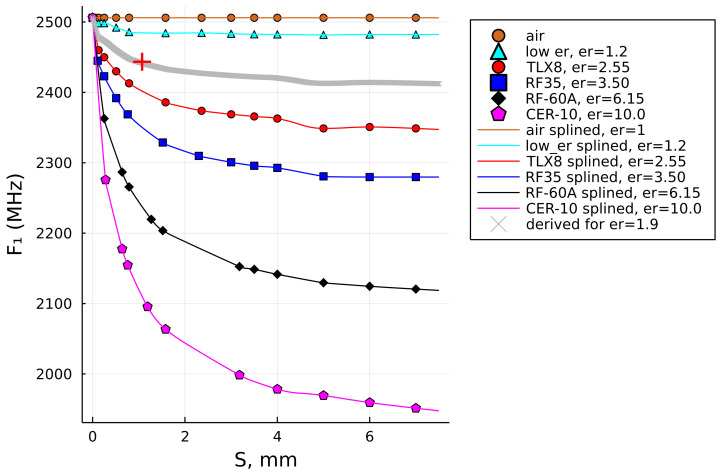
An example of how an additional reference curve was inserted to eventually improve the accuracy of the solution in a segment. A “lower” curve with $\epsilon_{r}=1.2$ (in cyan color) is inserted between the “air” curve and the “TLX8” curve. The outcome is that the computed value of the jeans SUT (here indicated by the red plus sign on the grey curve) is now 1.9, which is closer to the nominal value than before when it was equal to 2.0.

In general, we could say that a larger number of reference data points and curves enables more accurate solutions in finding the unknown $\epsilon_{x}$. Initially, one could start with the initial (Table 1) or extended (Table 3) reference data set and then append it with additional reference points according to the specific measurement interest.

### 6.2. A Comparison to Other Works

It is practically impossible to find references that would enable us directly compare our test results with theirs due to the difference in the permittivity values and thicknesses of the materials that were used, but the closest possible comparison with multiple references will be done in the paragraphs that follow.

For example, in [41], a double split-ring resonator (DSSR) was used on an FR4 substrate at the nominal frequency of 2.45 GHz. The authors claimed an error above 5% and only two solid materials were tested: silicon with a 5.92% error and rubber with 6.81% (see [41], Table 2), while in their Table 4, where the comparison with some other works was presented, the errors in those other works ranged from 1.30% in [42] and 2.26% in [43] to 6.89% in [44], for solid dielectrics. Another question involved is whether in ([41], Equation (1)), the quadratic polynomial fitting equation was applicable to some other frequencies outside the interval where the fitting equation was defined.

In [25], a complementary split-ring resonator (CSSR) was used and, unlike many other papers, the sample thickness was also included in the observation (which mattered for the result). However, the samples tested in ([25], Table II) were not thicker than 2 mm, while in our work here, the samples were tested up to 7.37 mm and the respective results are discussed. The method proposed in [25] was tested on (only) four standard laminates and the reported measurement errors were 2.27%, 7.60%, 0.45%, and 3.43%, which averaged an error of 3.44%.

In [45], a symmetrical split-ring resonator (SSRR) with two variants of spurline filters was proposed at 2.22 GHz. For merely three standard laminates, the declared accuracy was 97% to 98% (i.e., less than 3% error). The more accurate result exhibiting the reported error of less than 2% was achieved with the more sophisticated structure of a double-spurline filter, while the single-spurline filter variant exhibited an error of up to 5.45% in the case of the Rogers 5880 laminate. More comparisons can be found in ([45], Table II), reading that in [46], in spite of a very subtle design of their multiple split-ring resonator at 5 GHz, the error for the Rogers 5880 sample was 12.73%, while for FR4, it was 1.59%.

Lastly, in [47], a Minkowski fractal microstrip cavity resonator was designed at 5.8 GHz and tested on four materials. The error spanned from 1.72% for the Rogers 4350 to 4.37% for the “glass“ sample, with which it was not specified how thick that sample of glass was or how thick the “PVC” sample was and which source of information in those cases the authors specifically referred to, which would be more helpful for a better comparison.

In the preceding paragraphs, we exploited the results from the works that used different variants of a split-ring resonator (SRR). SRRs are generally considered more sensitive and, therefore, more accurate than just a (single) microstrip ring resonator. Keeping in mind that in this paper we did not focus on the design of the MRR, but rather on the algorithm that uses the results obtained by an MRR, it is then understood that if our results are comparable or better than the results obtained by an SRR, then the algorithm we are proposing here works quite well.

It is also noteworthy that all of the above references benchmarked their algorithms on no more than four samples, and in most of the references, it was not specified how thick the particular samples were, which is also relevant information for a fair comparison. Namely, saying that a sample is “standard“ is not unambiguous because it is known that various *standard* laminates are offered in multiple standard thicknesses that a customer can choose, such as a Rogers 5880 laminate [6] that was used as the benchmark sample in several works cited above. A similar case is with the FR4, which is possibly the most widely used dielectric. Although considered a *standard laminate*, its $\epsilon_{r}$ value is far from standard and we acknowledged it in Table 2. For example, in [41], its value was declared as 4.07, in [25,45] it was 4.4, while in [47] it was declared as 4.6. (In the case of our FR4 sample, the value was also taken as 4.6.) Therefore, for a fair comparison, one would have to use the particular FR4 sample that somebody else tested because its permittivity generally depends on the particular manufacturer; its value cannot just be copied from some other publication.

In contrast to the above-cited works (and many others), we tested our algorithm on as many as 11 samples of both standard and non-standard materials, with their thicknesses ranging from 0.80 to 7.37 mm, because we wanted to push it to the approximate limits of the applicability of this algorithm; for such a broad range of samples, we reported the maximum error to be below 10%. It is impossible to know how many samples the authors in the previous works actually tested and then presented only up to four of the most successful results. In particular, if we had confined our presented results up to the four most impressive results—namely, Taconic RF60A-0620, FR4, Plexiglass 2, and Plexiglass 3 (i.e., the latter two having different sample thicknesses and calculating the error based on $\epsilon_{r}=2.65$, which was the declared value of the PVC (PVC being a very similar material to Plexiglass) sample in [47]), then the respective errors in our case would be 0.81%, 2.17%, 3.77%, and 0% (see Table 2 and Table 4 within this work), with an average error of 1.69%, which is better than many earlier papers based on the same number of four presented samples. However, it was more relevant to us to faithfully and transparently disseminate the complete testing experience of ours, rather than just select a small number of the most impressive measurement results. In the interest of a fair comparison of the results, we just showed how the presented error values can be easily adjusted to sound more impressive, as the reader cannot really know how many samples had actually been tested by an author and there is no common standard on the minimum number and type of the sample materials that should be tested for a fair comparison between different papers.

## 7. Conclusions

In this paper, an efficient algorithm for the evaluation of relative permittivity of an unknown dielectric sheet material was described, tested, and compared against the 11 sample materials and the results obtained by a previously verified VMA algorithm. The proposed method showed good agreement with both the VMA and the reference permittivity values of the SUTs that were used for the benchmarking. The error was within 10% for the worst case (of the 7.37 mm thick SUT), the average error for all the presented samples was 5.31%, while the average error for the four standard laminates (i.e., TLX8-060, RF60A-0300, RF60A-0620, and FR4) was 2.34%. This was a satisfactory achievement given the fact that the input data for the algorithm were based on the measurements by a simple MRR. At the same time, the proposed RDSA is about 10 times faster than the VMA. The reference data set of the proposed RDSA is easily appendable with additional reference points if higher accuracy is desired, and is also frequency-scalable to be applicable to various MRR circuits. With an adequate adjustment, we believe it could be applied to work with other measurement methods, which serve to determine the dielectric or magnetic parameters of the SUT.

## Figures and Tables

**Figure 1 sensors-22-05591-f001:**
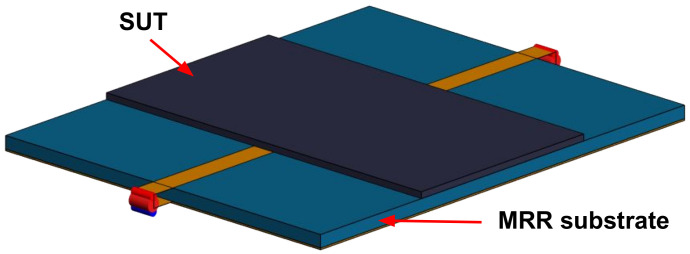
A computational model of an MRR with a SUT placed over it (e.g., TLX-8 is the SUT and FR4 is the substrate). The structure was modeled using the Feko full-wave solver [37].

**Figure 2 sensors-22-05591-f002:**
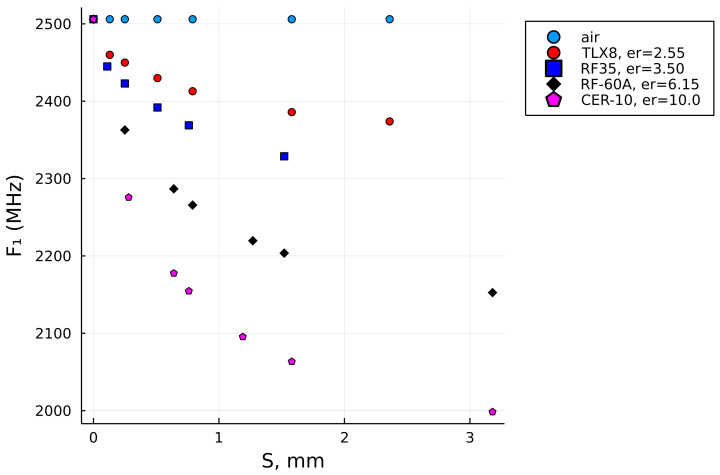
The resonant frequency values evaluated using a full-wave solver for the thickness- and permittivity values of the four known laminates.

**Figure 3 sensors-22-05591-f003:**
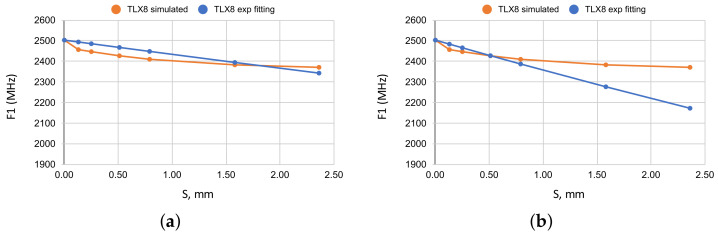
Attempts of exponential fitting of one set of the reference data. (**a**) TLX8: fitting with α=0.028 (better fitting for the last part of the curve). (**b**) TLX8: fitting with α=0.06 (better fitting for the first part of the curve).

**Figure 4 sensors-22-05591-f004:**
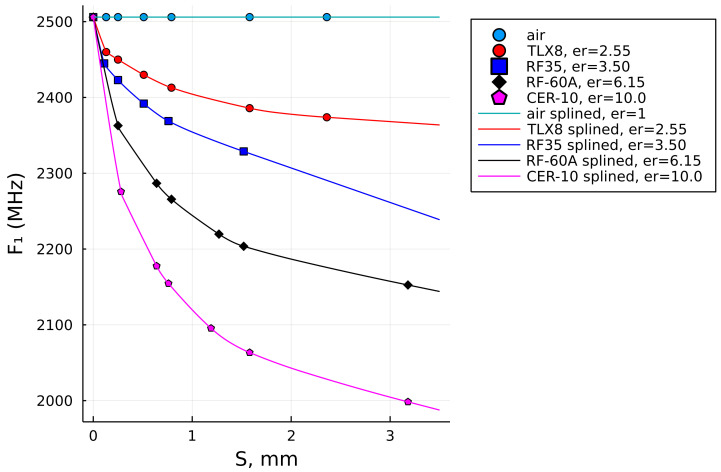
The reference points interpolated by a Schumaker spline shape-preserving algorithm in Julia programming language.

**Figure 5 sensors-22-05591-f005:**
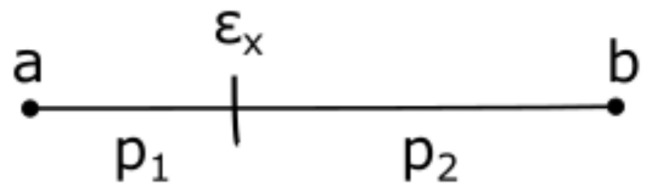
A horizontally-oriented illustration of the interval between any two bounding reference curves and the given (*S*, *F*1) point between them.

**Figure 6 sensors-22-05591-f006:**
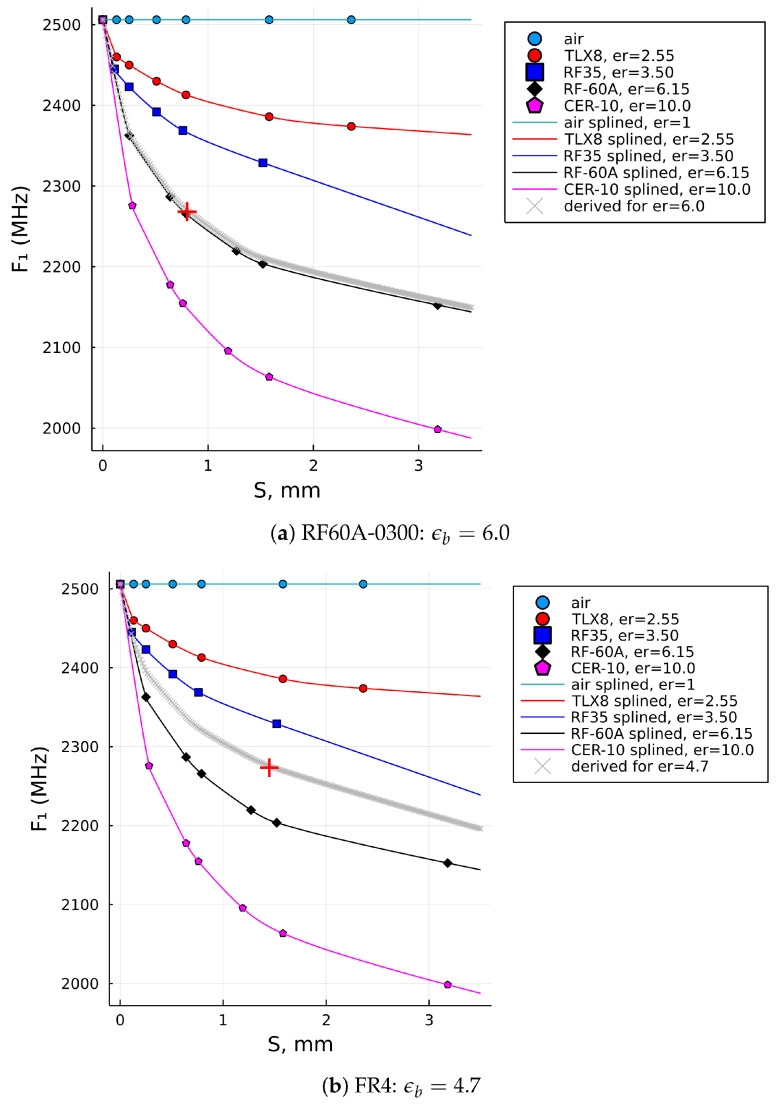
SUT permittivities computed by the proposed algorithm (**a**) for the RF60A-0300 laminate, (**b**) for the FR4 laminate, (**c**) for glass, and (**d**) for plexiglass 1.

**Figure 7 sensors-22-05591-f007:**
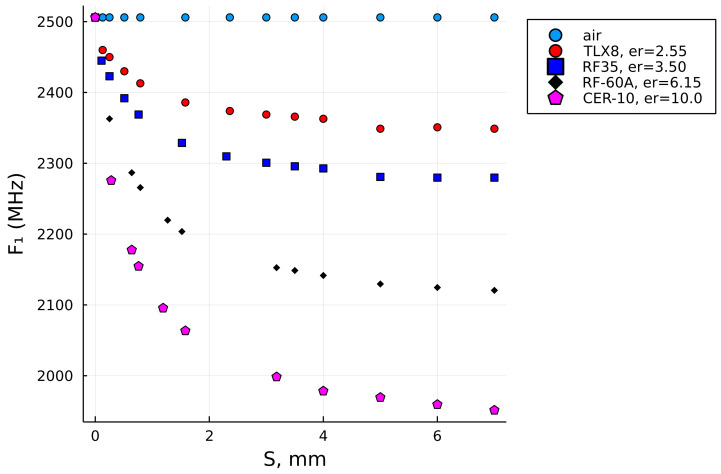
The extended set of reference points.

**Figure 8 sensors-22-05591-f008:**
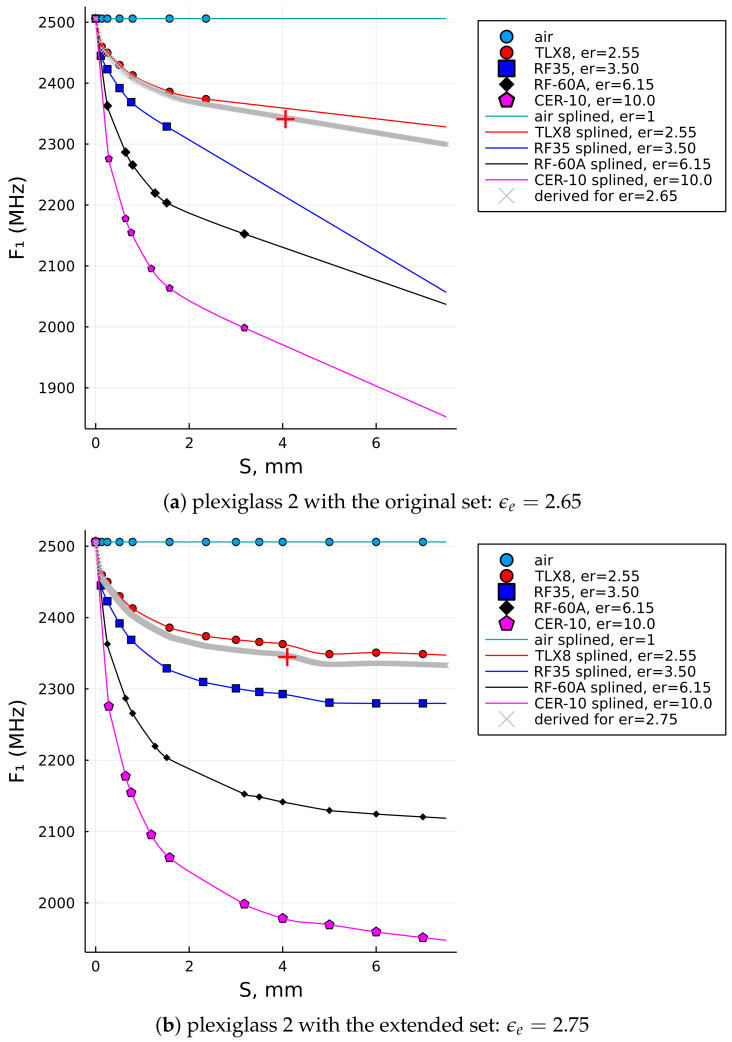
A comparison of the interpolation with the original reference data set (S≤3.18mm) vs. the extended reference data set (S≤7.5mm) and the solution for the plexiglass 2 sample with such settings. (**a**) the original reference data set. (**b**) the extended reference data set.

**Figure 9 sensors-22-05591-f009:**
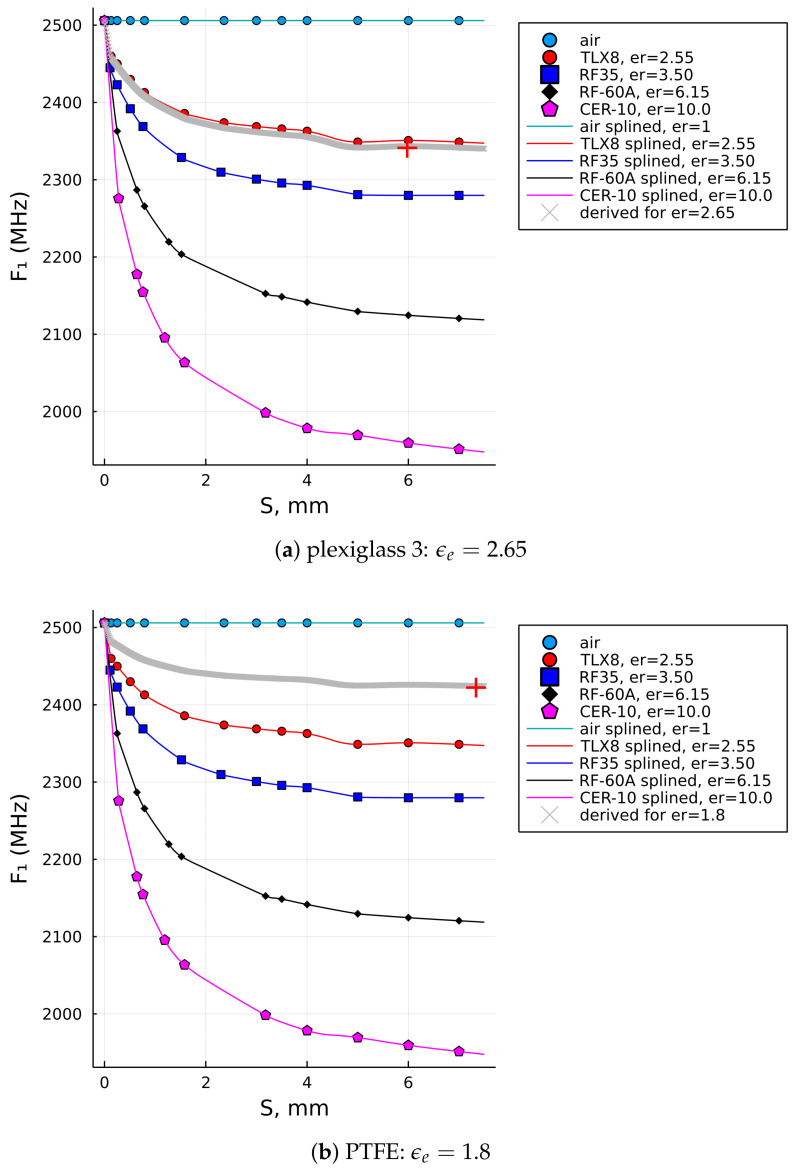
SUT permittivities computed by the proposed algorithm for plexiglass 3 and PTFE samples: (**a**) for plexiglass 3. (**b**) for PTFE.

**Table 1 sensors-22-05591-t001:** The reference data set.

Reference Laminate	Nominal Permittivity, $\epsilon_{n}$	Laminate Thickness, *S* (mm)					
TLX-8	2.55	0.13	0.25	0.51	0.79	1.58	2.36
RF-35	3.50	0.11	0.25	0.51	0.76	1.52	n/a
RF-60A	6.15	0.25	0.64	0.79	1.27	1.52	3.18
CER-10	10.0	0.28	0.64	0.76	1.19	1.58	3.18

**Table 2 sensors-22-05591-t002:** Testing this reference data set-based algorithm (RDSA) against known permittivities and the variational method-based algorithm (VMA) from [34].

SUT	S (mm)	$\epsilon_{n}$	$F_{1}$ (MHz) ${}^{*}$	$\epsilon_{x}$ by Alg. in ([34], Table 7)	$\epsilon_{b}$ by This Alg.
TLX8	0.8	2.55	2405	2.74	2.75
TLX8-060	1.57	2.55	2380	2.57	2.65
RF60A-0300	0.83	6.15	2268	6.2	6.0
RF60A-0620	1.59	6.15	2205	5.84	6.1
jeans	0.9	1.7	2447	1.86	2.0 ${}^{**}$
FR4	1.48	4.3–4.6	2276	4.44	4.7
glass	1.93	4–7	2166	6.28	6.85
Plexiglass 1	2.95	2.6–3.5	2375	2.34	2.5

* with: $F_{0}=2506\;\mathrm{MHz}$. ** See the discussion in Section 6.1 for means to further improve the result to 1.87.

**Table 3 sensors-22-05591-t003:** The reference data set extended by additional thickness points.

Reference Laminate	Nominal Permittivity, $\epsilon_{n}$	Laminate Thickness, *S* (mm)												
		Initial Data Set						Extended Data Set						
TLX-8	2.55	0.13	0.25	0.51	0.79	1.58	2.36	3.00	3.50	4.00	5.00	6.00	7.00	
RF-35	3.50	0.11	0.25	0.51	0.76	1.52	n/a	2.30	3.00	3.50	4.00	5.00	6.00	7.00
RF-60A	6.15	0.25	0.64	0.79	1.27	1.52	3.18	3.50	4.00	5.00	6.00	7.00		
CER-10	10.0	0.28	0.64	0.76	1.19	1.58	3.18	4.00	5.00	6.00	7.00			

**Table 4 sensors-22-05591-t004:** Benchmarking three thicker samples using the RDSA proposed here.

SUT	S (mm)	$\epsilon_{n}$	$F_{1}$ (MHz)${}^{*}$	$\epsilon_{x}$ by alg. in ([34], Table 7)	$\epsilon_{b}$ by this alg. (orig. data)	$\epsilon_{e}$ by this alg. (ext. data)
Plexiglass 2	4.13	2.6–3.5	2349	2.57	2.65	2.75
Plexiglass 3	6	2.6–3.5	2349	2.49	2.5	2.65
PTFE	7.37	2	2428	1.67	1.7	1.8

* with: $F_{0}=2506\;\mathrm{MHz}$.

## Data Availability

Not applicable.

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
