# Peer review of "An Efficient and Frequency-Scalable Algorithm for the Evaluation of Relative Permittivity Based on a Reference Data Set and a Microstrip Ring Resonator"

_sensors, 2022, doi:10.3390/s22155591_

Round 1
Reviewer 1 Report
Please see attachment.

Reviewer 2 Report
The paper is interesting and method suggested is useful. Main problem is English. It should be enhanced.
Reviewer 3 Report
This manuscript is on the investigation of an efficient algorithm for evaluating relative permittivity of an unknown dielectric sheet material at microwave bands. Time cost of same computing task and the preparation of the reference data set are compared with another method and discussed, the experimental results matched perfectly with the designed performance. This piece of research work can be a very good evaluating method for other research groups in exploring the EM properties of materials. This manuscript can be accepted for publication after the minor revision with the author addressed the following issues:
1. The format of some formulas and Table 2 should be adjusted and aligned, and the resolution of figs should be improved for higher readability.
2. Could we get the dielectric and thickness properties of the magnetic materials through this method?
3. The evaluation error of the permittivity and thickness in this work is about 10%, which is not enough within some occasions. And how to further improve the test accuracy?
4. The dispersion effect cannot be ignored for evaluating the permittivity of materials, what is the applicable frequency range for this method and how to adjust it?
5. Some effictive paremeters retriving methods for artificial metamaterials have been reported elsewhere, e.g., Physical review E, 2005, 71(3): 036617, Optics Express, 2011, 19(18): 17413-17420, and Scientific Reports, 2015, 5:8434, which may help enrich the introduction.
